# Aging and Thymosin Alpha-1

**DOI:** 10.3390/ijms262311470

**Published:** 2025-11-27

**Authors:** Maria A. Simonova, Igor Ivanov, Natalia S. Shoshina, Alina M. Komyakova, Dmitry A. Makarov, Denis S. Baranovskii, Ilya D. Klabukov, Kristina P. Telepenina, Dmitrii A. Atiakshin, Peter V. Shegay, Andrey D. Kaprin, Vasiliy N. Stepanenko

**Affiliations:** 1Shemyakin-Ovchinnikov Institute of Bioorganic Chemistry of the Russian Academy of Sciences (IBCh RAS), Ul.Miklukho-Maklaya 16/10, 117997 Moscow, Russia; 2Lomonosov Institute of Fine Chemical Technologies, MIREA—Russian Technological University, Vernadskogo pr. 86, 119571 Moscow, Russia; ivanov_i@mirea.ru; 3Center for Synthetic Biotechnology, Sechenov University, 119991 Moscow, Russia; nats_lee@mail.ru (N.S.S.); komyakova.1999@mail.ru (A.M.K.); youngchemist@mail.ru (D.A.M.); telepenina_kristina@mail.ru (K.P.T.); stepanenko_v_n@staff.sechenov.ru (V.N.S.); 4Department of Regenerative Medicine, National Medical Research Radiological Centre of the Ministry of Health of the Russian Federation, Koroleva St., 249036 Obninsk, Russia; denis.baranovskii@unibas.ch (D.S.B.); ilya.klabukov@gmail.com (I.D.K.); dr.shegai@mail.ru (P.V.S.); kaprin@mail.ru (A.D.K.); 5Institute for Systems Biology and Medicine, Russian University of Medicine, Delegatskaya Str. 20/1, 127473 Moscow, Russia; 6University Hospital Basel, Basel University, 4001 Basel, Switzerland; 7Obninsk Institute for Nuclear Power Engineering, National Research Nuclear University MEPhI, 249020 Obninsk, Russia; 8Scientific and Educational Resource Center for Innovative Technologies of Immunophenotyping, Digital Spatial Profiling and Ultrastructural Analysis, Patrice Lumumba Peoples’ Friendship University of Russia (RUDN University), 117198 Moscow, Russia; atyakshin-da@rudn.ru

**Keywords:** aging, thymus, immunosenescence, *Refnot*, rejuvenation, senescence, thymosin alpha-1, TNF-alpha

## Abstract

Aging is characterized by immune decline, mainly due to thymic involution—the age-related shrinkage of the thymus gland. This leads to reduced T-cell production, chronic inflammation, and increased susceptibility to age-related diseases. Thymosin alpha-1 (Tα1), a peptide hormone produced by the thymus, exhibits potent immunomodulatory, anti-inflammatory, and antioxidant properties. It helps restore immune function by stimulating T-cell differentiation, enhancing thymic output, and modulating dendritic cell and macrophage activity. Preclinical and clinical studies show that Tα1 can improve vaccine response in the elderly and mitigate immunosenescence. The hybrid drug *Refnot* (a fusion of tumor necrosis factor alpha (TNFα) and Tα1) combines Tα1’s immunomodulation with TNF’s antitumor activity but has reduced toxicity. It represents a promising therapeutic approach to counteract age-related immune dysfunction and inflammation, potentially by slowing the aging process. Further research is needed to validate its long-term efficacy and safety in geriatrics.

## 1. Introduction

Aging is a complex biological process that results in a gradual deterioration of body functions associated with DNA damage, epigenetic changes, cellular dysfunction and chronic inflammation. Modern science is actively seeking ways to slow down or reverse the aging process.

The thymus is a primary lymphoid organ responsible for generating T lymphocytes. It is located in the upper front part of the chest, just in front of the heart. The thymus consists of two lobes, each containing a central medulla and an outer cortex. The thymus gland plays a key role in aging, since its age-related involution (decrease and replacement with fat tissue) is directly associated with a weakening of the immune system, chronic inflammation and an increased risk of age-related diseases. Thymosins are a group of peptide hormones produced by the thymus gland that play a crucial role in regulating the immune system and influencing the aging process. Their effect on aging is associated with the maintenance of T-lymphocyte function, anti-inflammatory action and protection against age-related diseases. Thymosin-α1 (Tα1), one of the thymosins, exhibits a broad spectrum of biological activities, including immunomodulation and anti-inflammatory effects, and has potential as a therapeutic agent for combating aging. This review focuses on the role of thymosin-α1 (Tα1) in the aging process, as well as the potential therapeutic applications of the recombinant hybrid protein combining tumor necrosis factor (TNF) and thymosin-α1 (*Refnot*).

## 2. The Role of the Thymus in Aging

The thymus reaches its peak activity during childhood, but its function declines after puberty due to its mass reduction and the replacement of functional tissue with fat. Age-related atrophy of the thymus gland, which is commonly named thymus involution, is a natural process involving the reduction in the volume and functional activity of the thymus. This process has critical consequences for the immune system. The rate of thymus involution varies individually, leading some researchers to propose that the thymus serves as an endogenous determinant of both general (within a species) and individual lifespan [1].

Physiological involution of the thymus is characterized morphologically by a progressive and continuous decline in the number of immature cortical lymphocytes, accompanied by a structural restructuring of the organ (Figure 1).

The main signs of involution are a decline in the number of cortical, immunologically immature thymocytes to 95%; disruption of the lobular structure of the thymus, leading to formation of the so-called inverted thymus, a mixture of cortical and cerebral areas; and a significant decrease in the population of stromal cells within the thymic microenvironment, along with the loss of their regulatory functions. However, the complete disappearance of the functional units of the thymus does not occur. The cells of the reticuloepithelial (RE) network, along with associated non-lymphatic stromal elements—such as dendritic cells, macrophages, Langerhans cells, and interdigitating cells—and the remaining population of cortical thymocytes are preserved [2,7].

Being one of the central organs of the immune system, the thymus performs the following functions: (i) controls proliferation, differentiation, selection and final maturation of T lymphocytes; (ii) produces thymic and non-thymic hormones that affect the functions of T lymphocytes and other cells [1].

In the thymus, early T-cell precursors differentiate into double negative (DN) (CD4^−^CD8^−^) cells, which are further divided into the subpopulations DN1 (CD44^+^CD25^−^), DN2 (CD44^+^CD25^+^), DN3 (CD44^−^CD25^+^), and DN4 (CD44^−^CD25^−^). Specific cell markers like CD44 and CD25 influence aging by reflecting the accumulation of senescent and memory immune cells. The naive cells express a low level of CD44, whereas in memory cells, the expression level of CD44 is high [8]. Naive T cells are generally CD25-negative, as they have not been activated by an antigen. CD25 is an early activation marker and a high-affinity receptor for IL-2, which is crucial for the proliferation and function of certain immune cells. CD25 expression is primarily seen on activated T cells and on regulatory T cells (Tregs) [9]. With age, there is a decrease in naive T cells (CD44^−^ CD25^−^) and an increase in memory cells [10], like CD44^+^ or regulatory T cells CD4^+^CD25^+^ [11]. This shift leads to immune dysregulation, reduced immune responses, and chronic inflammation

Generally, the DN precursors differentiate into double positive (DP) (CD4^+^CD8^+^) cells, which, through positive and negative selection, develop into single positive (SP) (CD4^+^CD8^−^ or CD4^−^CD8^+^) mature naive T cells. With age, the frequency of DN subpopulations increases, while that of DP subpopulations decreases. At the same time, a decline in the DN2 and DN3 subpopulations is observed within the DN precursors. Additionally, with age, CD3^+^ DN precursors accumulate in the thymus. Aging also affects the late stages of T-cell development. In old age, DP and SP thymocytes exhibit disrupted regulation of CD3 expression, which may lead to weakened TCR-dependent stimulation of thymocytes. Thymocytes show a reduced response to mitogens, which is evidenced by the loss of expression of the activation marker CD69 and a lack of proliferation. The content of T-cell receptor excision circles (TREC), which serve as markers for newly produced T cells, also decreases in the thymus with age. These changes can result in reduced production of naive T cells in the body and a significant narrowing of the T-cell receptor (TCR) repertoire. As a result, homeostatic expansion of existing T cells may occur, with memory T cells becoming the predominant population. Disturbances in immune regulation caused by decreased thymus activity may also contribute to inflammatory or chronic conditions [12].

The process of thymic involution is primarily attributed to the degeneration of thymic stromal cells, which undergo functional changes with age [4]. Stromal cells, unlike lymphocytes, produce very low levels of catalase. This deficiency leads to an accumulation of hydrogen peroxide and increased oxidative stress in stromal cells, contributing to thymic atrophy.

Several molecular mechanisms can slow down the age-related degeneration of thymus functions. The molecules that can be affected in this process include transcription factors (Foxn1, E2F3, myc), small interfering RNA (miR-181a-5p), Wnt signaling pathway molecules (Wnt4), growth factors (FGF21, IGF-1, KGF), cytokines (IL-7, IL-22), hormones and receptors (leptin, somatotropin, ghrelin, GHSR) and others (lamin-B1). Among the molecules that promote thymus involution are small interfering RNA miR-125a-5p, miR-205-5p, cytokines IL-6, SCF, IL-1β, LIF, Wnt signaling pathway molecules (Axin) and others (OSM, follistatin) [12]. Sex hormones also play a key role in thymus involution. Thus, endogenous androgens have potent effects on the thymus, causing atrophy [13], whereas androgen treatment in vitro causes DP thymocyte apoptosis largely mediated through induction of DP TNFα production [14].

In addition to its primary role in the maturation of T lymphocytes, the thymus also functions as part of the endocrine system. The thymus produces its own (thymic) hormones, as well as hormones that are mainly produced by other endocrine glands. The non-thymic hormones produced in the thymus are listed in Table 1. In the aging organism, thymic hormones may modulate age-related processes and help to maintain cognitive abilities and memory. Their positive effect on the sex glands, pituitary gland, and the hypothalamic–pituitary–gonadal axis as a whole has been shown. Biologically active glucocorticoids (GCs) are also synthesized in the thymus, and glucocorticoid receptors are expressed in thymocytes. The expression of some neuropeptides, like somatostatin, substance P, vasoactive intestinal peptide, calcitonin gene-related peptide, and neuropeptide Y. Prolactin, luteinizing hormone, follicle-stimulating hormone (FSH), corticotropin-releasing factor, oxytocin, and vasopressin are also produced by thymus cells, and the corresponding receptors are expressed in cells. The expression of other non-thymic hormones, such as insulin, has been shown in the thymus, and the corresponding receptors have been found on various thymus cells. These non-thymic hormones produced can be transported by migrating T cells [1].

During thymic involution progress, the associated endocrine interactions may become disrupted and thus, contribute to cellular aging and promote the development of age-related diseases.

## 3. The Role of Thymosin-α1 in the Aging Process

Thymosin-α1 (Tα1) is the first biologically active peptide isolated from fraction 5 of thymosin, a calf thymus extract. Its molecular weight is 3108 Da, and its length comprises 28 amino acids. The N-terminal amino group of Tα1 is acetylated. Tα1 is formed by the cleavage of the prothymosin-α (PTMA) (Figure 2) by asparagine endopeptidase. Although PTMA contributes to cell cycle regulation and apoptosis, its role in aging is multifaceted. PTMA is a multifunctional, small acidic protein that plays a crucial role in regulating cell death and survival pathways. The PTMA gene is linked to aging through its influence on cell proliferation and survival—processes that are essential for tissue repair and affected by age. Several studies have shown that PTMA acts as a negative regulator of apoptosis [51] by disrupting apoptosome formation and inhibiting caspase-9 activation. In certain contexts, PTMA partners with proteins such as p8 to form anti-apoptotic complexes, while some chemical agents can counteract its inhibitory effects, thereby restoring apoptosis. Additionally, PTMA directly interacts with Keap1 [52], releasing transcription factor Nrf-2 to induce the expression of genes that combat oxidative stress. Its phosphorylation status is also critical for its cytoprotective and anti-apoptotic activities [53].

Although specific receptors for Tα1 have not been identified so far, it is a pleiotropic hormone that interacts with different protein molecules. In a neutral solution, Tα1 has a disordered structure, but at lower pH, either in the presence of lipid membranes or when interacting with other proteins, it adopts a partially ordered conformation. Due to the acetylated N-terminus, Tα1 can anchor in membranes and interact with receptors, modulating various signaling cascades. Tα1 belongs to intrinsically disordered proteins (peptides), which are more sensitive to environmental changes like salt concentration, pH, temperature, metabolites, etc., when compared to other structured proteins. It likely functions as a “sensor” of the cellular environment. The biological effects of Tα1 include immunomodulation and antitumor, antiviral and anti-inflammatory activity, but it is also involved in other physiological processes [55,56].

Due to the absence of toxicity and owing to its wide spectrum of biological action, Tα1 is used in clinical practice as an immunomodulatory drug. Commercially available Tα1, which is known as *Thymalfasin (Zadaxin)*, is produced by total chemical synthesis. Genetic engineering expression makes use of the advances in biotechnology to produce purified recombinant Tα1 [57]. *Thymalfasin* has been approved in 35 countries and is used in the treatment of viral hepatitis B and C, human immunodeficiency virus (HIV) infection, COVID-19, various oncological diseases, bacterial and fungal infections, and autoimmune diseases [58,59] and is used as an adjuvant for influenza vaccines [58]. A study of gene expression in human peripheral blood mononuclear cells has shown that Tα1 increases the expression of 1198 out of 8300 genes [60]. These genes are associated with energy metabolism, protein and DNA synthesis, regulation of the cell cycle and apoptosis, intracellular signaling, etc. It is likely that Tα1 compensates for the age-related decline in expression of certain genes, thereby potentially slowing down the aging process [61,62]. This points to a possible role for Tα1 in cellular homeostasis and age-related regulation of gene activity.

The main aspects of Tα1 biological activity in the context of mitigating the negative effects associated with aging will be discussed below.

### 3.1. Immunomodulatory Activity

Tα1 can have a significant impact on the aging process, mainly due to its activity in the thymus and peripheral immune system. Tα1 increases the expression of the IL-7 gene in peripheral blood mononuclear cells [60]. IL-7 is critical for the survival and differentiation of thymocytes at the initial and late stages of their maturation and also maintains the functional activity of mature peripheral T cells [63]. Tα1 stimulated the secretion of IL-7 during co-cultivation of stem cells with thymic epithelium, enhancing thymopoiesis—the proliferation and differentiation of stem cells into T cells [64]. Tα1 prevents the death of thymocytes induced by various agents [65,66], as well as reduces the level of reactive oxygen species in thymocytes [65]. Tα1’s protective effect depends on the activation of protein kinase C (PKC) [65,66]. Another possible mechanism of thymocyte protection from apoptosis is inhibition of thymocyte interaction with galectin-1 (Gal-1) by Tα1. Gal-1 is known to promote apoptosis of thymocytes and peripheral T cells [67,68,69]. Tα1 binds to Gal-1, modulating its interaction with lactose and suppressing the biological activity of Gal-1 [70]. Tα1-Gal-1 interaction can protect thymocytes and peripheral T cells from apoptosis, maintaining immune homeostasis. By acting through multiple mechanisms, Tα1 can stimulate the thymus to produce new T lymphocytes, whose numbers naturally decline with age. In addition to its effect on the thymus, Tα1 enhances bone marrow cell proliferation, especially in the presence of various stimulating factors (MCSF, GMCSF, GCSF, and IL-3) [71]. The abovementioned Tα1 activity may slow down the age-related decline in the number of new immune cells. Tα1 modulates the functions of the peripheral immune system, primarily by targeting dendritic cells [72,73,74,75]. It enhances their activation, maturation, and differentiation from bone marrow cells under the control of GMCSF/IL-4, and this process depends on the innate immunity receptors TLR9 and IFN-αβR [76,77]. In dendritic cells, Tα1 enhances phagocytosis [72], expression of HLA-DR molecules (key for antigen presentation) [75,76], expression of costimulatory molecules CD40, CD80 [74,75,76] and CD86 [72,76], necessary for T-lymphocyte activation. Tα1 reduces the expression of PD-1L (an inhibitor of T-cell activity), which can contribute to an enhanced immune response, and increases the expression of TIM-3 (a factor in the exhaustion of CD8^+^ T lymphocytes) [74], which indicates a dual regulatory role of Tα1. Under the influence of Tα1, dendritic cells increase the secretion of the anti-inflammatory cytokine IL-10 and the T-cell stimulating factor IL-12 in response to various pathogens [72,77]. Activated by Tα1, dendritic cells promote the proliferation and functional activity of regulatory CD4^+^FoxP3^+^ T lymphocytes (Treg) [75,77,78], which play a key role in suppressing excessive inflammation and maintaining immune tolerance.

Thus, by modulating the activity of dendritic cells and enhancing the function of Treg, Tα1:Reduces age-related inflammation by increasing IL-10 levels and suppressing immune hyperreactivity;Maintains immune homeostasis, preventing both excessive activation (autoimmune reactions) and immune deficiency;Improves T-cell function, compensating for the age-related decline in their activity;Corrects the imbalance between immune activation and tolerance, which is especially important in atherosclerosis, neurodegeneration, and metabolic disorders.

Tα1 also has a complex effect on the innate immune system, modulating the activity of dendritic cells and macrophages/monocytes. First, it has been shown that activation of Tα1 dendritic cells depends on TLR9 receptors [72,77,79], IFN-αβR [77], and the enzyme indolylamine 2,3-deoxygenase-1 (IDO-1) [77]. Tα1 increases IDO-1 levels in dendritic cells through TLR9- and IFN-αβR receptor-dependent pathways, which promotes Treg activation [77,80], enhances synthesis of the anti-inflammatory cytokine IL-10, and reduces formation of proinflammatory cytokines IL-1β and IL-17A [80]. Tα1 upregulates the expression of TLR2, TLR5, and TLR9 [72] and coactivates the TLR9/MyD88-dependent antiviral pathway, stimulating the production of IFNα and IL-12 [72,79].

In macrophages, Tα1 enhances phagocytosis [81], reduces the production of proinflammatory cytokines (TNFα, IL-1β, IL-6) [82], and increases the level of the anti-inflammatory cytokine IL-10 [83,84]. Tα1 has been shown to suppress lipopolysaccharide (LPS)-induced inflammation in macrophages by reducing the expression of NF-κB, AP-1, TLR4, and p53, increasing the level of Nrf-2 protein (oxidative stress protection factor), and inhibiting the SAPK/JNK and NF-κB signaling pathways [82]. Thus, Tα1 may exert a “rejuvenating” effect on the immune system by suppressing age-related inflammation and enhancing the function of macrophages and dendritic cells.

### 3.2. Antioxidant Activity

Tα1 reduces the formation of reactive oxygen species in various cells: macrophages [82], thymocytes [65], splenocytes [85], and astrocytes [86]. Antioxidant activity of Tα1 has also been demonstrated in vivo. In rats with induced steatohepatitis, Tα1 increased superoxide dismutase activity [87], and in rabbits with experimental atherosclerosis, it reduced lipid peroxidation levels [88]. Tα1 has also been shown to directly neutralize hydrogen peroxide and superoxide radicals in solutions [86]. Thus, Tα1 can have a direct effect by protecting cells from oxidative damage and preventing age-associated diseases like atherosclerosis, neurodegenerative disorders, and metabolic disorders. In particular, Tα1 can exert a protective effect on thymic epithelial cells, which are especially vulnerable to oxidative stress [12].

### 3.3. Activity in the Nervous System

Different changes in the Tα1 level during stress [89,90], and its localization in brain structures such as the hippocampus, astrocytes, and spinal neurons [91,92] may indicate its important role in the function of the nervous system. Tα1 closely interacts with the hypothalamic–pituitary system, stimulating the secretion of key pituitary hormones: adrenocorticotropic, thyrotropic and luteinizing [93]. At the same time, it suppresses the production of thyrotropin-releasing hormone, corticoliberin and somatostatin by hypothalamic cells [94], thus being important for maintaining neuroendocrine balance during aging. Tα1 has an anti-inflammatory effect in the nervous system, suppressing proinflammatory cytokines (TNFα, IFNγ) and activating neuroprotective pathways such as Wnt3a/β-catenin [95]. This is important in the context of age-related neuroinflammation, which contributes to the development of neurodegenerative diseases.

A decrease in prothymosin-α (the precursor of Tα1) expression in microglial cells under the influence of β-amyloid suggests its potential role in the pathogenesis of Alzheimer’s disease [96]. At the same time, Tα1 stimulates neurogenesis and improves cognitive functions [97], which makes it a promising candidate for the treatment of age-related cognitive impairment. An important aspect is also the ability of Tα1 to restore the levels of nerve growth factor (NGF) and its receptor p75NGFr in the brain after thymectomy [98]. In addition, Tα1 modulates synaptic transmission in the hippocampus [99], which may affect neuronal plasticity and cognitive function in old age. The beneficial effect of Tα1 on the human nervous system was demonstrated in a clinical study. In patients with common variable immunodeficiency complicated by major depressive disorder, Tα1 administration not only stabilized immunological parameters but also alleviated depressive symptoms [100].

### 3.4. Activity in the Cardiovascular System

Tα1 exhibits inhibitory activity against angiotensin-converting enzyme (ACE) [86,101] and reduces ACE2 expression, decreasing the synthesis of angiotensin 1–7 [101] in pulmonary epithelial cells. By inhibiting ACE, Tα1 can limit excessive activation of the renin–angiotensin system, which theoretically helps to reduce blood pressure.

Experimental studies using a rabbit atherosclerosis model have demonstrated that Tα1 significantly decreases lipid levels in both blood plasma and erythrocytes. In parallel, an increase in the activity of erythrocyte Na+/K+-ATPase, a key enzyme that maintains ionic homeostasis of cells, was recorded [102]. Thus, Tα1 may have anti-atherosclerotic activity and maintain membrane homeostasis.

### 3.5. Age-Related Changes in Tα1 Production and Their Consequences

The main source of Tα1 in the body is the epithelial cells of the thymus, primarily subcapsular cortical and medullary cells [103,104,105,106,107]. With age, the ability of the thymus to produce Tα1 [106,108] and its precursor protein prothymosin-α [109,110] declines, most likely due to a reduction in the number of epithelial cells. It has also been demonstrated that Tα1 production decreases in subcapsular cortical cells during accidental thymic involution, thus correlating with the progression of involution [111].

Age-related changes in the level of Tα1 in blood have not been sufficiently studied. There is evidence that its concentration in blood plasma decreases with age [112], but another study did not reveal significant changes [113]. Inflammatory and autoimmune diseases are characterized by a reduced concentration of Tα1 in blood plasma [114]. Taking into account the above-described physiological effects, the age-related decline in Tα1 production may be an important aspect of aging and may have the following consequences:Decreased thymic function;Impaired immune surveillance;Imbalance between immune activation and tolerance;Chronic inflammation;Decreased regulatory influence on the hypothalamic–pituitary system;Decreased neuroprotective potential;Oxidative stress.

The data on the age-related decline in Tα1 production emphasize its important role in the aging process and the need for further research to develop new strategies to compensate for this deficit in old age.

### 3.6. Biological Activity of Tα1 and Aging: Research Data

To date, there are a limited number of studies demonstrating the effects of Tα1 on the immune system of elderly organisms. These effects are listed in Table 2.

Tα1 demonstrated the ability to compensate for age-related decline in immune function, including restoration of T-cell responses and enhancement of humoral responses. Thus, Tα1 is a pleiotropic hormone capable of modulating key aspects of immune aging. Further research should be aimed at optimizing its use in geriatric practice, including combinations with other geriatric protective agents.

## 4. Potential Applications of *Refnot* for Addressing Age-Related Changes

The drug *Refnot*, which was developed in Russia, is a recombinant protein (TNF-T) combining the N-terminal sequence identical to TNFα, a linker of three amino acids (Ile-Asp-Met), and a C-terminal sequence identical to Tα1. *Refnot* is used as an immunomodulating and antitumor agent in cancer patients [126,127,128,129]. TNF-T exhibits unique pharmacological properties due to its hybrid structure. The biological activity of TNF-T is similar to the activity of Tα1. TNF-T exhibits adjuvant activity during vaccination by activating the T-cell link of immunity [130]. Both TNF-T and Tα1 dose-dependently enhance the expression of MHC I, CD4, and CD8 on thymocytes; however, the activity profiles for CD4 are different. Thus, Tα1 exhibits a monotonic stimulatory effect without inhibitory activity, whereas TNF-T induces a biphasic response—enhancing expression at low doses but suppressing it at high doses—and additionally inhibits thymocyte proliferation, an effect not observed with Tα1. Like Tα1, TNF-T stimulates the proliferation of splenocytes and lymph node cells [126] and exhibits a number of activities of TNFα. It induces the death of cells sensitive to TNFα, but its toxicity is reduced by a factor of 10 when compared to TNFα. Like TNFα, TNF-T activates phagocytosis in macrophages but without the inhibitory effect at high doses observed in the case of TNFα. TNF-T and TNFα potentiate the antiviral activity of IFNα against the vesicular stomatitis virus (VSV), while the effectiveness of TNF-T is one order of magnitude higher than that of TNFα. Both proteins modulate the expression of MHC I, CD4 and CD8 on thymocytes, but the dose-dependent profiles of TNF-T and TNFα differ significantly. TNF-T demonstrates a wider range of effective stimulating concentrations [126]. Finally, a unique activity characteristic of TNF-T is its ability to stimulate the cytotoxic activity of natural killer cells [126].

As mentioned above, one of the main targets of Tα1 are dendritic cells. There are no data so far on the activity of TNF-T on these cells, but it is known that Tα1 and TNFα, either individually or in combination, enhance their maturation and differentiation, while Tα1 inhibits TNFα-induced IL-12 production [76]. Clinical observations in cancer patients provide evidence supporting the immunomodulatory potential of TNF-T. It has a positive effect on both T lymphocytes and natural killer cells. An increase or, in some cases, normalization of the number of T lymphocytes, CD4^+^ and CD8^+^ cells, as well as normalization of the CD4^+^/CD8^+^ ratio, was recorded. TNF-T also increased the number and activity of natural killer cells [126,131,132]. TNF-T demonstrates a significantly better safety profile compared to TNFα. Its systemic toxicity, which is reduced by two orders of magnitude, is confirmed by good tolerability in clinical studies [126]. This advantage correlates with pharmacokinetic differences—the half-life (T½) of TNF-T is 210–280 min, which is 7–9 times longer than T½ of TNFα (15–30 min [133,134]) and 40–100% longer than that of Tα1 (120 min [135]).

Thus, owing to its hybrid structure, *Refnot* combines immunomodulatory activity (both Tα1 and TNFα), direct antitumor action (due to TNFα and, possibly, Tα1), and synergism with interferon, which can enhance antiviral protection. In view of the abovementioned results, it is likely that *Refnot* can influence key mechanisms of aging such as chronic inflammation and immune dysfunction (immunosenescence), both of which accompany the inflammaging states [136].

Considering that TNF-T enhances macrophage phagocytosis and that the Tα1/TNFα combination demonstrates immunomodulatory activity against dendritic cells, it can be inferred that TNF-T has the potential to mitigate inflammatory processes, including those associated with aging and inflammatory tissue damage.

Decreased adaptive immunity (T and B cells) in old age increases susceptibility of organisms to infections and cancer. Given the immunomodulatory effects of TNF-T on thymocytes, T cells and NK cells, as well as its ability to enhance the antiviral activity of IFNγ, it has the potential to slow down immunosenescence while enhancing antiviral immunity. The additional antiangiogenic and cytostatic activity of TNF-T may also contribute to the prevention of age-associated tumor progression.

As mentioned above, the spectrum of biological activities of TNF-T partially overlaps with that of Tα1. This similarity suggests that TNF-T may have additional beneficial effects that are significant in the context of aging, including antioxidant and neuroprotective activity or cardioprotective properties. Instrumental studies (PET-CT, CT, or MRI) have repeatedly shown that the use of *Refnot* in patients aged 55 and older is accompanied by signs of thymus activation, which in turn leads to the activation of T lymphocytes. These cells of the immune system are responsible for protecting the body from pathogens (viruses, bacteria) and abnormal cells (cancer or senescent cells).

Compared to other pharmacological strategies for aging therapy, *Refnot* may offer several advantages, as it has minor side effects. For instance, mTOR inhibitors, while being explored for their potential to slow aging, have been observed to increase the risk of hyperglycemia and hyperlipidemia. The relationship between mTOR and aging is complex; while its overactivation is linked to aging, chronic inhibition can also lead to negative metabolic consequences [137]. Similarly, senolytics (senotherapy) can cause different side effects like gastrointestinal discomfort and cardiovascular problems. Other potential issues include skin conditions and hematologic toxicities. A significant challenge is the risk of inducing drug resistance or the unintended re-initiation of cell division in previously senescent cells that could promote cancer [138,139].

## 5. Conclusions

The thymus plays a crucial role in the production of immune cells, but its function irreversibly declines with age. This decline forms the basis of the immunological theory of aging. Increasing the level of T lymphocytes in the blood is critically important for restoring the function of the immune system, its “rejuvenation” and, as a result, supporting comprehensive body health. This approach potentially contributes to the effective removal of damaged, mutated cells, abnormal proteins, DNA, and pathogens. In this way, *Refnot* may have an anti-aging potential. However, additional experimental or clinical data, as well as evaluation of its combinations with other geroprotectors such as rapamycin and metformin, are required to directly support its effectiveness in treating age-related pathologies.

## Figures and Tables

**Figure 1 ijms-26-11470-f001:**
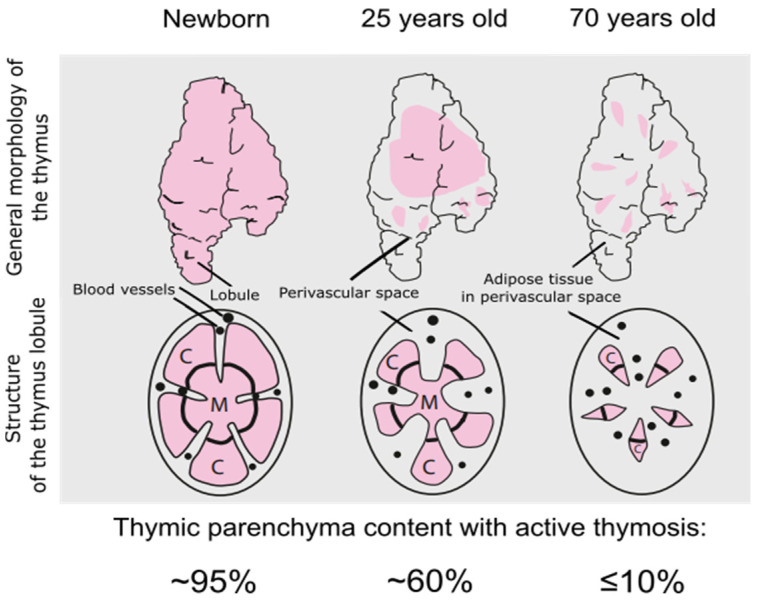
A schematic representation of age-related thymic involution, adapted from [2,3]. C—cortex, M—medulla. In aging, the thymus parenchyma is gradually replaced by fat and connective tissue in the perivascular spaces (**upper panels**) [4]. The outermost layer, the capsule, thickens with age and extends into the gland via septa, but these septa themselves may become thin and filled by adipocytes [5]. The sharp demarcation between the cortex and medulla becomes less clear and disorganized (**lower panels**) [6]. Images were created using Inkscape v1.4.2 software.

**Figure 2 ijms-26-11470-f002:**
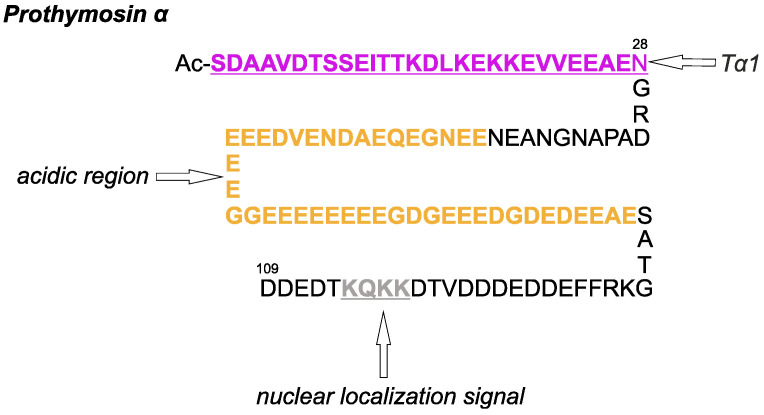
Human prothymosin-α amino acid sequence (1–109) [54]. N-terminal serine is acetylated (Ac). Tα1 sequence is indicated in purple, acidic region is indicated in orange, and nuclear localization signal is shown in gray.

**Table 1 ijms-26-11470-t001:** Non-self (atopic) hormones of the thymus and their possible roles in aging.

Hormone	Role in Aging	Activities in Thymus	Refs.
Glucocorticoids (GCs)	Accelerate immunosenescence; cause bone and muscle loss	Dual role: induce apoptosis and support thymocyte survival	[15,16,17]
Somatostatin	Decline linked to cognitive impairment and sleep disturbances	Decreased receptor levels implicated in thymic involution	[18,19]
Substance P	Decline impairs tissue repair and nerve function; anti-inflammatory	Protects thymocytes from apoptosis	[20,21]
Vasointestinal peptide	Regulates inflammation, cell survival, and circadian rhythm	Protects thymocytes from apoptosis; promotes T-cell differentiation	[22,23,24,25]
Calcitonin gene-related peptide	Decline impairs bone and vascular health	Solely inhibitory: induces apoptosis and blocks T-cell development	[26,27]
Neuropeptide Y	Promotes autophagy, which declines with age	Promotes thymocyte proliferation (in young animals)	[28,29]
Growth hormone (GH)	Maintains muscle mass, bone density, and metabolism	Stimulates thymocyte/TEC proliferation, thymic secretion, and T-cell export	[30,31]
Prolactin	Mitigates retinal dysfunction and neuronal injury	Induces thymulin production in TECs	[1,32,33]
Luteinizing hormone	Increased levels affect the brain	Enhances the proliferation of thymocytes	[1,34]
Follicle-stimulating hormone (FSH)	Contributes to fat accumulation and bone loss	Unclear activity in thymus	[35]
Corticotropin-releasing factor	Contributes to age-related decline	Unclear activity in thymus	[36,37]
Oxytocin	Improves muscle, bone, and brain function	Unclear activity in thymus	[38,39,40]
Vasopressin	Altered secretion leads to cardiovascular and kidney issues	Unclear activity in thymus	[41]
Melatonin	Antioxidant; regulates sleep–wake cycle	Increases thymus weight and elevates thymic hormone levels	[42,43,44]
Insulin	High levels may accelerate aging; crucial for brain function	Alters thymocyte synthesis; absence leads to autoimmune diabetes	[45,46,47]
Insulin-like growth factor (IGF)	Decline associated with aging; disrupted signaling extends lifespan	Unclear activity in thymus	[48,49]
Calcitonin	Decline contributes to bone loss (post-menopause)	Unclear activity in thymus	[50]

**Table 2 ijms-26-11470-t002:** Physiological effects of Tα1 in elderly and thymus-deficient organisms.

Species, Age	Condition or Pathology	Tα1 Effect	Ref.
Mice 15–24 months	Immunodeficiency, aging	Stimulation of T-helper activity	[115,116,117]
Mice 9–20 months	Aging	Increased T-cell precursor counts	[118]
Mice 23 months	Aging	Enhanced antibody production following immunization with T-dependent (tetanus toxoid) but not T-independent antigen (pneumococcal capsular polysaccharide)	[119]
Athymic mice	Athymic	Induction of CD90.2 expression (a marker of mature T cells) on null lymphocytes (T or B cell markers)	[120]
Mice a model of thymus involution under the influence of hydrocortisone	Thymus involution	Enhancement of the proliferative response of splenocytes and thymocytes to IL-1, IL-2, a mixture of cytokines, concavalin A and phytohemagglutinin	[121]
Rhesus macaques 18–25 years old	Aging	Increased activity of natural killer cells	[122]
Humans 65–92 years old	Aging, influenza vaccination	Increased synthesis of specific antibodies by peripheral blood mononuclear cells after influenza vaccination	[123]
Humans 65–99 years old	Aging, influenza vaccination	Restoration of the humoral immune response to influenza vaccine	[124]
Humans 65 years and older	Aging, influenza vaccination	Increase in specific antibody titer after influenza vaccination and mitigation of side effects	[124]
Humans 65 years and older	Aging, COVID-19 vaccination	Correlation between plasma Tα1 level and the presence of IgG antibodies against SARS-CoV-2 S protein after COVID-19 vaccination: the duration of protection depends on the Tα1 concentration	[125]

## Data Availability

No new data were created or analyzed in this study. Data sharing is not applicable to this article.

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
