# Peer review of "Aging and Thymosin Alpha-1"

_ijms, 2025, doi:10.3390/ijms262311470_

Round 1
Reviewer 1 Report
Comments and Suggestions for Authors
The manuscript is well written and contains a lot of valuable information. However, I have a few comments that need to be addressed. The authors mention that "somatostatin, substance P, vasoactive intestinal peptide, calcitonin gene-related peptide, neuropeptide Y, prolactin, luteinizing hormone, follicle-stimulating hormone, corticotropin-releasing factor, oxytocin, and vasopressin" are produced by thymus cells. It would be helpful if the authors could provide a table listing each hormone or steroid produced by the thymus along with its role in aging.
Additionally, it would be interesting to highlight genes associated with aging that are linked to increases in Thymosin-α1 under the section "The Role of Thymosin-α1 in the Aging Process."
Finally, it would be beneficial if the authors could explain how specific markers or cell populations such as CD44–CD25–positively or negatively influence aging.
Author Response
Comments 1: The authors mention that "somatostatin, substance P, vasoactive intestinal peptide, calcitonin gene-related peptide, neuropeptide Y, prolactin, luteinizing hormone, follicle-stimulating hormone, corticotropin-releasing factor, oxytocin, and vasopressin" are produced by thymus cells. It would be helpful if the authors could provide a table listing each hormone or steroid produced by the thymus along with its role in aging.
Response 1: Following the advice of the reviewer we have added additional table (new Table 1).
Comments 2: Additionally, it would be interesting to highlight genes associated with aging that are linked to increases in Thymosin-α1 under the section "The Role of Thymosin-α1 in the Aging Process."
Response 2: Following the advice of the reviewer we introduced a short paragraph (lines 166-177) which clarify linage of PTMA gene to aging.
Comments 3: Finally, it would be beneficial if the authors could explain how specific markers or cell populations such as CD44–CD25–positively or negatively influence aging.
Response 3: Following advice of the reviewer we introduced an additional paragraph (lines 96-105) to clarify how specific markers (cell populations CD44–CD25–) negatively influence aging.
Reviewer 2 Report
Comments and Suggestions for Authors
1.The application of Refnot in anti-aging remains speculative. Although the article suggests that Refnot may have anti-aging potential, there is very limited experimental or clinical data that directly supports its application in the elderly population. It is more based on inferences from the known functions of Tα1 and TNFα.
2.Some details in the mechanism diagram are not clear enough. Although the changes in "thymic lobule structure" and "thymic parenchyma content" in Figure 1 are intuitive, they lack quantitative data support and do not explain the source of the images or whether they have undergone statistical analysis.
3.Lack of comparison and discussion of other anti-aging strategies: The article does not compare Tα1/Refnot with current mainstream anti-aging interventions (such as rapamycin, NAD+ supplements, senolytics, etc.), and lacks in-depth analysis of its position in the field of anti-aging.
Author Response
Comment 1: The application of Refnot in anti-aging remains speculative. Although the article suggests that Refnot may have anti-aging potential, there is very limited experimental or clinical data that directly supports its application in the elderly population. It is more based on inferences from the known functions of Tα1 and TNFα.
Response 1: We thank the reviewer for his valuable comment. Indeed, there are very limited experimental or clinical data obtained so far and we addressed this issue in the conclusions section.
Comment 2: Some details in the mechanism diagram are not clear enough. Although the changes in "thymic lobule structure" and "thymic parenchyma content" in Figure 1 are intuitive, they lack quantitative data support and do not explain the source of the images or whether they have undergone statistical analysis.
Response 2: Following the advice of the reviewer we modified the legend to the figure 1 addressing this issue and provided appropriate literature references.
Comment 3: Lack of comparison and discussion of other anti-aging strategies: The article does not compare Tα1/Refnot with current mainstream anti-aging interventions (such as rapamycin, NAD+ supplements, senolytics, etc.), and lacks in-depth analysis of its position in the field of anti-aging.
Response 3: Following the advice of the reviewer we introduced the discussion of the strategies in the context of side effects (lines 416-426).
Reviewer 3 Report
Comments and Suggestions for Authors
This review article provides a comprehensive and overview of the role of thymic involution in immunosensecence and the potential therapeutic applications of Thymosin alpha-1 (Tα1) and the hybrid drug Refnot (TNF-T) in counteracting age-related immune decline. Moreover, the manuscript logically progresses from the problem (thymic involution) to a potential solution (Tα1/Refnot) and particularly, it excellently details the molecular and cellular mechanisms of Tα1, covering immunomodulatory, antioxidant, neuroprotective, and cardiovascular activities.
This paper a valuable and informative review that synthesizes a large body of literature. With the incorporation of the suggested revisions, it has the potential to be an outstanding contribution to the field.
Some major points for consideration and corrections:
- I consider the introduction to be highly appropriate. However, when discussing the thymus, it would be advisable to include a brief anatomical description of the gland, as well as its location within the body.
- Figure 1 is very suitable for understanding the morphoanatomy of the thymus. Nevertheless, a detailed description of this figure should be included within the body of the article, with a direct reference to the image, to enable a more specific understanding of its characteristics.
- At the end of line 86, where the role of the thymus as a central organ of the immune system is concluded, the corresponding citation crediting this section is missing.
- In line No. 120, it is stated: "Sex hormones also play a key role in thymus involution." In what specific way might they intervene in thymic involution?
- In the section "The Role of Thymosin-α1" (line No. 138), I consider it appropriate to include a figure that allows for the visualization of this molecule (Thymosin-α1) to better understand its chemical characteristics (predominant amino acids, bonds, and peptide structure), rather than solely discussing its molecular weight and the 28 amino acids that comprise it. Furthermore, a visual representation of Tα1's structure and a schematic summarizing its pleiotropic effects would greatly enhance reader comprehension.
- In lines 155-156, where it is mentioned that Tα1 is used in clinical practice, what are the specific doses, concentrations, or units of this Tα1 that are employed?
- Regarding Table 1, I recommend that a column should be added to the table specifying under which circumstances or pathologies Tα1 led to the improvement, stimulation, or increase in immune or biochemical modulation. This is currently unclear, particularly when referring to studies in mice or Rhesus macaques. I recommend restructuring the table for a more comprehensive description, it requires more context (e.g., the experimental model/disease state) to be fully interpretable.
Author Response
Comments 1: I consider the introduction to be highly appropriate. However, when discussing the thymus, it would be advisable to include a brief anatomical description of the gland, as well as its location within the body.
Response 1: Following the advice of the reviewer we introduced a brief description of the gland, as well as its location within the body (lines 47-49).
Comments 2: Figure 1 is very suitable for understanding the morphoanatomy of the thymus. Nevertheless, a detailed description of this figure should be included within the body of the article, with a direct reference to the image, to enable a more specific understanding of its characteristics.
Response 2: Following the advice of the reviewer we modified the legend for figure 1 accordingly.
Comments 3: At the end of line 86, where the role of the thymus as a central organ of the immune system is concluded, the corresponding citation crediting this section is missing.
Response 3: Following the advice of the reviewer we have added the citation (line 93 in revised manuscript).
Comments 4: In line No. 120, it is stated: "Sex hormones also play a key role in thymus involution." In what specific way might they intervene in thymic involution?
Response 4: Following the advice of the reviewer we modified accordingly and added missing information. Now it reads: Sex hormones also play a key role in thymus involution . Thus, endogenous androgens have potent effects on the thymus causing atrophy ([13], 10.3389/fimmu.2018.00794), whereas androgen treatment in vitro causes DP thymocyte apoptosis largely mediated through induction of DP TNFα production ([14],10.4049/jimmunol.164.4.1689).
Comments 5: In the section "The Role of Thymosin-α1" (line No. 138), I consider it appropriate to include a figure that allows for the visualization of this molecule (Thymosin-α1) to better understand its chemical characteristics (predominant amino acids, bonds, and peptide structure), rather than solely discussing its molecular weight and the 28 amino acids that comprise it. Furthermore, a visual representation of Tα1's structure and a schematic summarizing its pleiotropic effects would greatly enhance reader comprehension.
Response 5: Following the advice of the reviewer we created the figure 2 to better understand chemical characteristics of Thymosin-α1 and its precursor.
Comments 6: In lines 155-156, where it is mentioned that Tα1 is used in clinical practice, what are the specific doses, concentrations, or units of this Tα1 that are employed?
Response 6: Since this review targets a broad readership, clinical practice details—available elsewhere—are intentionally omitted for clarity.
Comments 7: Regarding Table 1, I recommend that a column should be added to the table specifying under which circumstances or pathologies Tα1 led to the improvement, stimulation, or increase in immune or biochemical modulation. This is currently unclear, particularly when referring to studies in mice or Rhesus macaques. I recommend restructuring the table for a more comprehensive description, it requires more context (e.g., the experimental model/disease state) to be fully interpretable.
Response 7: Following the advice of the reviewer we have added the additional column to table and changed the table number to Table 2.
Round 2
Reviewer 3 Report
Comments and Suggestions for Authors
I would like to inform you that I have no further comments regarding your manuscript. I sincerely appreciate the time and effort you have dedicated to addressing the observations provided.